# Particle Deposition in the Vicinity of Multiple Film Cooling Holes

**DOI:** 10.3390/mi13040523

**Published:** 2022-03-26

**Authors:** Yubo Peng, Guoqiang Xu, Xiang Luo, Jian He, Dongdong Liu

**Affiliations:** National Key Laboratory of Science and Technology on Aero-Engine Aero-Thermodynamics, The Collaborative Innovation Center for Advanced Aero-Engine of China, Beihang University, Beijing 100191, China; dream070141@buaa.edu.cn (Y.P.); 04822@buaa.edu.cn (G.X.); clhejian@126.com (J.H.)

**Keywords:** film cooling, heat transfer, particle deposition

## Abstract

Particle deposition on film cooling surface is an engineering issue that degrades the thermal protection of turbine blade. Here, we present a combined experimental and numerical investigation on the particle deposition in the vicinity of multiple film cooling holes to reveal the effect of interactions between cooling outflows on particle deposition. The numerical simulation of film cooling with a group of three rows of straight film cooling holes is conducted and validated by experimental data with blowing ratios ranging from 0 to 0.08. Wax particles with size range from 5 to 40 μm are added in the heated mainstream to simulate the particle deposition in the experiment. The simulation results show the decrease of particle deposition with blowing ratio and various deposition characteristics in different regions of the surface. The flow fields from numerical results are analyzed in detail to illustrate deposition mechanism of the particles in different regions under the interactions of cooling outflows. The cooling air from the holes in the first row reduces the particle concentration near the wall but causes particle deposition in or between the tail regions by the generated flow disturbance. The cooling air from the latter hole separates the diluted flow in the upstream from the wall, and creates a tail region without particle deposition. This revealed particle deposition characteristics under the effect of outflows interaction can benefit the understanding of particle deposition in engineering applications, where multi-row of cooling holes are utilized.

## 1. Introduction

For thermal protection of the turbine blade, cooling air from secondary air system is supplied through the elaborately arranged film cooling holes on the blade to form a thin cooling air film for reduction of heat transfer from the hot mainstream flow. This so-called film cooling technology has been widely used for decades in turbomachinery for its high reliability in turbine blade cooling. Numerous studies have been carried out to reveal the mechanism of film cooling, optimize the design of hole, and improve cooling efficiency [1,2,3,4,5,6].

One of the challenges facing in film cooling is the deposition of particles, which are mainly originated from the pollutants, e.g., sand, debris, and ash, in inlet air or fuel, on the turbine blade [7,8,9]. These deposits can easily change the surface morphology and block the cooling holes, resulting in the deviation of the film cooling from its original design and thus degrading of the film cooling efficiency and endangering the safety of the turbine blade. To obtain further insights into the characteristics of particle deposition on different areas of the turbine blade, the particle deposition was divided into the external surface deposition and internal wall deposition.

For external particle deposition, Demling et al. [10] presented the effects of deposition accumulation in the upstream and downstream of the film hole on the film cooling efficiency. Ai et al. [11] focused on the deposition of pollutants in coal derived syngas near the film cooling hole. It was found that the deposition mainly occurred in the upstream of the film hole and the downstream area between the two film holes. However, there was less deposition coverage in the downstream of the hole. Somawardhana et al. [12] investigated the effect of geometric parameters on the cooling efficiency and found that the deposition shape had little effect on the cooling efficiency, but the deposition height had a great impact on the cooling efficiency. For the internal hole blockage, Pan et al. [13] studied the effect of particle blockage in the monoclinic hole on film cooling, and analyzed the influence of different positions of the blockage in the hole on film cooling. The results showed that the blockage can severely affect the film cooling when it was on the upstream surface of the hole’s inner wall. Cardwell et al. [14] studied the effect of different particle blockages on the mass flow of cooling air under a specific pressure ratio. Wolff et al. [9] explored the impact of particle size on the deposition on the film cooling. The results showed that the particle with smaller size was easier to deposit, but the large particle was likely to destroy the deposition. Barker et al. [15] experimentally studied the influence of hole diameter, hole spacing, and hole inclination on hole blockage, and found that the hole inclination was the most important parameter in affecting the blockage inside the hole. Moreover, some researchers [16] studied the effects of both the external deposition and internal blockage on film cooling.

Many studies also focused on the dynamics of the deposition process in film cooling by changing the flow parameters or hole geometry. Rozati et al. [17] simulated the deposition of particles at the leading edge of film cooling blades, and the results indicated that the increase of blowing ratio reduced the deposition. Ai et al. [18] studied the effects of blowing ratio on deposition growth and blade surface temperature. It was found that the increase of blowing ratio led to the decrease of blade surface temperature and the growth of deposition. Lawson et al. [19] experimentally investigated the effect of particle deposition on film cooling of single row inclined holes. The results illustrated that with the increase of film cooling momentum ratio, the reduction of film cooling efficiency caused by particle deposition had a peak. Mensch et al. [20] investigated the particle deposition on the blade end wall with vapor film cooling. The experiment result revealed that the particle deposition had both positive and negative effects on the wall heat transfer. On one hand, the deposition was helpful to the heat insulation. On the other hand, the deposition changed the wall roughness and increased the wall heat transfer. For the geometry of the film cooling hole, Wang et al. [21] studied the effects of different film hole shapes on particle deposition. Wang et al. [22] numerically simulated the film cooling of single row laidback fan-shaped cooling holes to study the interaction between particles and cooling wall.

Researches on the particle deposition were mostly limited to the configuration of a single hole or a single row of holes, neglecting the interactions between the outflows from holes in different rows, which may play an important role in practical applications, as multiple rows of film cooling holes were utilized. This multiple row configuration aims not only for approaching the reality, but also for simulating the transpiration cooling, the extreme form of film cooling, by using the straight holes [23,24]. In this study, we utilize a multi-row configuration, three rows of staggered straight holes to both simulate the transpiration cooling in engineering application and simplify the geometry factor, to investigate the effect of outflow interactions on particle deposition characteristics on flat surface. Commercial software Fluent and its own discrete phase model are applied to numerically simulate the particle deposition process. Experiment of particle deposition using wax particle is conducted to compare with the simulation data. The calculation reveals how the interactions of the outflows from different cooling holes affect the particle deposition characteristics on a flat surface.

## 2. Materials and Methods

### 2.1. Experimental Apparatus

A low speed open loop wind tunnel is utilized for the experiment of the particle deposition. The schematic of the experimental setup is shown in Figure 1. An air compressor supplies the air flow, including the flow in the mainstream and the cooling air. These two flows are controlled by two gas control valves separately. One FCI-ST98 thermal flowmeter with the uncertainty of ±0.5% is assembled in each of these two flow channels to measure the flow rate. After the flowmeter, the air in the mainstream is heated by an electric heater to temperature $T_{g}$ of 102 ${}^{\circ }$C. After heating, melted wax with melting temperature at 44 ${}^{\circ }$C supplied from a pressurized container, is sprayed into the mainstream channel via a spray nozzle, which gives a particle size range between 5 and 40 μm. The particles fully mix with the hot air and flow into the rectangular channel of test section, which is 30 mm wide and 45 mm high. For the cooling air, it is led into a coolant chamber for pressure stabilization, in which its temperature $T_{c}$ and pressure $P_{c}$ are measured. A stainless plate with 204 staggered arranged film cooling holes is assembled between the coolant chamber and mainstream channel, allowing the cooling air to flow through the cooling hole into the mainstream. The duration of the particle deposition experiment is 30 s. In the outlet of the tunnel, a filter is placed to collect the added wax particles. In the experiment, the weight of the wax container and film cooling plate are measured before and after the experiment to calculate the weight of used wax and deposited wax on the surface, respectively. Two calibrated K type thermocouples are fixed in the flow channels to monitor the temperature of the cooling air in the cooling chamber and hot air in the mainstream.

The blowing ratio *F*, which measures the mass flow ratio between the cooling air and hot air in the mainstream [25], can be defined as follows
(1)$$F=\frac{\rho_{c}u_{c}}{\rho_{g}u_{g}},$$
where $\rho_{c}$ and $\rho_{g}$ are the densities of the coolant and main flow, respectively, and $u_{c}$ and $u_{g}$ respectively, are the velocities of the coolant in the cooling chamber and main flow. Based on the parameters in engineering applications, we vary the blowing ratio *F* from 0 to 0.08 for both experiment and simulation. The cooling efficiency η [6] is defined as follows
(2)$$\eta =\frac{T_{aw}-T_{w}}{T_{aw}-T_{c}},$$
where $T_{aw}$ is the adiabatic wall temperature, $T_{w}$ the surface temperature, and $T_{c}$ the cooling air temperature.

### 2.2. Particle Deposition Simulation

In numerical simulation, we choose a representative unit, including seven straight film cooling holes, one in the center and the other six surrounding holes arranged in the corners of a hexagon. As illustrated in Figure 2a, the simulation model consists of a coolant chamber and a mainstream channel with a width of 9 mm along the *x* axis, a height of 15 mm along the *z* axis, and a length of 49 mm along *y* axis. Seven straight film cooling holes connect the coolant chamber and mainstream channel. The diameter of the cooling hole is 1 mm, and the distances between the holes in the flow direction *y* and the transverse direction *x* are both 3 mm, same as the experiment.

The commercial software ANSYS ICEM is used to generate Hexahedral mesh in the fluid domain. O-grid mesh is applied in the region of film cooling holes for high mesh quality. The detail of the mesh is shown in Figure 2b,c. Mesh refinement is conducted near the wall to keep the Yplus smaller than 1.2. The total quantity of elements in the mesh is 3.55 million. Fluent is chosen for numerical simulation in this study for its convenience in modeling the particle deposition. We use the idea gas as the working fluid due to the low flowing velocity and small range of temperature in this study. For the boundary condition, the inlet velocity in the mainstream is kept at 20 m/s and the temperature is set at 475 K. All the walls are set as non-slippery and adiabatic surfaces. The mass flow rate of the cooling gas with inlet temperature of 300 K is adjusted according to the blowing ratio *F* of the case. Since this calculation intends to simulate the particle deposition in transpiration cooling, therefore the outflow angle between the film cooling hole and y-axis is set at 90${}^{\circ }$, and the blowing ratio *F* is varied between 0 and 0.08—the same as in transpiration cooling.

To calculate the particle deposition and track the particle in the mainstream channel, discrete phase model (DPM) is implemented upon the flow field and heat transfer simulation. The trajectory of the particle is calculated using the Lagrangian method, which tracks the motion of each particle in the flow field. The motion of the particle is mainly determined by the flow field, but is also affected by its intrinsic random motion. To capture the motion of the particles and avoid the difficulties in coupled simulation caused by the interaction between the particles and flow field, several assumptions are made: (1) each particle is considered as a non-spinning hard sphere with uniform density; (2) the interaction between particles are neglected due to the low mass fraction (∼$10^{-6}$ kg/m^3^) of the particles in this study; (3) the added particles do not affect the flow field; (4) the rebound of the particle caused by the surface roughness is neglected. Based on these assumptions, the particle is considered to be trapped on the wall when it contacts with the surface. If the particle is not colliding with the surface, its motion will be calculated until it leaves the controlled volume. Therefore, we are able to simulate the process of particle deposition in the mainstream channel. It has to be noted that the simulation is a steady state simulation by setting the mass flow rate of the added particle at $3\times 10^{-8}$ kg/s for duration of 1 s.

In this steady state simulation, the initial velocity of the added particle is set at 0 m/s, since the initial velocity is found to barely affect the deposition result. Based on the experiment, the range of the particle size in calculation is between 5 and 40 μm, following the Rosin–Rammler distribution and having an average diameter of 10 μm. The physical properties of the added wax particle are shown in Table 1.

### 2.3. Simulation Validation

To choose a suitable turbulent model in the calculation of film cooling, we calculate the film cooling of a single hole based on the reference [26] using four turbulent models. The particle deposition is not considered in this simulation.

The geometry of the model is shown in Figure 3a. According to the settings in reference [26], we build a similar model with the mainstream channel having a length of 294 mm, a width of 18 mm, and a height of 90 mm. The diameter of the cooling hole is 6 mm and its inclination angle is 35${}^{\circ }$. The inlet velocity in the mainstream is set at 20 m/s, and the gas temperature at 298 K. The gas temperature in the cooling flow is kept at 188 K, and the blowing ratio is set at 0.08.

We present the simulation results, including the averaged cooling efficiency $\eta_{a}$ in transverse direction and the cooling efficiency in the center line along the flow direction, comparing it with the experiment data from reference [26] in Figure 3b,c. Except for the result calculated using the turbulent model SST k−ω, other simulation results have similar trends with the experimental data. Since the result from the turbulent model, RNG k−ε, is the best when comparing with the experimental data, we choose this turbulent model in the particle deposition simulation in this study.

### 2.4. The Initial Velocity of Added Particles

In this section, we explore the effect of the initial velocity $V_{p}$ of the added particle on the particle deposition on the surface. In the experiment, the particles are added into the mainstream via a spraying nozzle, which is difficult to simulate in the calculation. Therefore, we assume the particles are uniformly added into the mainstream with same initial velocity $V_{p}$.

We calculate three cases with same blowing ratio F=0.08 but change the initial velocity of the added particle from 0 to 20 m/s. The simulation results for different $V_{p}$ are presented in Figure 4. In general, the contours of deposited particle thickness *h* between three cases are qualitatively similar (Figure 4a–c). We also extract the deposition thickness along the *x* axis on the surface at y=1.5 mm, the particle mass concentration *c* along *x*-axis direction at (z=1 mm, y=1.5 mm) and (z=5 mm, y=1.5 mm) (Figure 4d–e). Although the thickness and concentration in some areas change, which can be attributed to the random motion of the particles, the data from three cases generally show insignificant differences, suggesting negligible effect of the particle’s initial velocity on the particle deposition simulation. Therefore, we use the result of $V_{p}=0$ for analysis of particle deposition in this study.

## 3. Results and Discussion

### 3.1. Particle Capture Efficiency

In the experiment, the particle deposition on the surface is a time-dependent process. The particles added in the mainstream would keep accumulating on the surface with time, resulting in an ever-increasing deposition thickness. However, we use steady state simulation for modeling the particle deposition. To validate our modeling result on particle deposition, we compare the particle capture efficiency ε, the ratio between the mass of the deposited particles on the surface and that of the added particles in the mainstream, from experiment and simulation, as shown in Figure 5.

The result in Figure 5 shows the monotonic decrease of the particle capture efficiency ε with blowing ratio *F* from both the experiment result and simulation data. The increase in *F* means more cooling air is supplied through the film cooling holes on the flat surface to prevent the particle deposition, leading to less deposition of the particles on the surface. The simulation result has a similar trend with the experimental data, although the simulation underestimates the particle deposition. The underestimation of ε from the simulation may be caused by the surface roughness on the wall or the effect of added particle on the mainstream flow, which are both not considered in the simulation. However, the comparison validates our simulation result of particle deposition on the surface.

### 3.2. The Effect of Blowing Ratio on Particle Deposition

For the added particle, due to its higher density than that of surrounding gas, it intends to fall upon the surface under the gravitational force. Although the particle’s small size and high gas velocity in the mainstream induce the added particles to follow the gas and move through the channel, the low velocity field in the boundary layer near the wall makes the particle deposition possible. When the blowing ratio is zero, i.e., no cooling air is supplied through the film cooling holes in the wall, the particles collide with the surface and deposit on the wall under the gravity (Figure 6a). With the increase of the flow distance and correspondingly the duration of falling particles, more particles are likely to contact the surface, inducing the increase of *h* in the flow direction. The irregular pattern of the deposited particles that appeared on the surface (Figure 6a) may be caused by the random motion of the particles. We measure the averaged deposition thickness $h_{a}$ in several 1 mm × 1 mm areas along the flow direction *y*-axis with the center points lying on the lines s1 and s2 (Figure 6b), and plot the $h_{a}$ with the *y* coordinate $y_{c}$ of the center point of the chosen areas in Figure 6c. The data show the increase of $h_{a}$ with $y_{c}$ in general. Here, we define a thickness $h_{0}$, the averaged deposition thickness ($h_{0}=128\;\mu$m) in the square region where −4.5 mm ≤y≤ 4.5 mm and −4.5 mm ≤x≤ 4.5 mm for F=0.

With the supply of the cooling air on the surface (F=0.02), the thickness *h* of the deposited particle substantially decreases (Figure 6a). The injected cooling air not only gives an upward momentum to the particles near the surface but also separates the particles from the surface, both preventing the particles from depositing on the surface. With the increase of blowing ratio *F*, the deposition thickness *h* generally decreases on the surface. For the convenience of the description, we denote the holes from “H1” to “H7”, and their tail regions, i.e., the regions behind the holes along the flow direction from “HT1” to “HT7”, as shown in Figure 6b. In particular, we observe some deposition patterns of the particles in the tail regions of the film cooling holes. For the tail regions in HT1 and HT2, the deposited thickness of particles are almost the same as the region in between these two regions when F=0.02 and F=0.08. For F=0.05, *h* in HT1 and HT2 becomes larger than the surrounding region. However, the thickness *h* in the tail regions behind other film cooling holes, from H3 to H7, are significantly smaller than the surrounding regions, particularly when F=0.08. These phenomena are also found in the experiment, as shown in Figure 6d.

To obtain a quantitative result of the particle deposition on the surface when F>0, we measure the $h_{a}$ in several 1 mm × 1 mm areas along s1 and s2, avoiding the film cooling holes, and normalize $h_{a}$ using $h_{0}$ (Figure 7). The variation of $h_{a}/h_{0}$ with $y_{c}$ on s1 and s2 are generally different. For $h_{a}/h_{0}$ on s1 (Figure 7a), we observe the increase of $h_{a}/h_{0}$ with $y_{c}$ before the film cooling hole H4 (where $y_{c}=0$), followed by the drop of $h_{a}/h_{0}$ around H4 and then the increase of $h_{a}/h_{0}$ with $y_{c}$. The drop of $h_{a}/h_{0}$ is attributed to the cooling flow from H4, which reduces the particle deposition, while the increase of $h_{a}/h_{0}$ after H4 may be induced by the reduced effect of the film cooling resulting from the detachment of cooling air from the surface. For $h_{a}/h_{0}$ on s2 (Figure 7b), the variation of $h_{a}/h_{0}$ with $y_{c}$ over two cooling holes, H2 and H7, are significantly different. The dimensionless thickness $h_{a}/h_{0}$ keeps increasing after the first cooling hole H2 (where $y_{c}=-3$ mm), and the drop of $h_{a}/h_{0}$ is only observed after the second hole H7 (where $y_{c}=3$ mm). The difference in the deposition characteristics after H4 and H2 could be resulted from the different incoming flow conditions of these two holes. For the flow before H2 on s2, it is undisturbed and can be considered as uniform; while for the flow before H4 on s1, the incoming flow field is already been affected by cooling outflow from H1 and H2. Therefore, the interaction between outflows from different film cooling holes results in different particle deposition characteristics, compared with that generated by one single film cooling outflow.

### 3.3. The Interaction between the Outflows of Multiple Film Cooling Holes

To obtain further insights into the particle deposition characteristics resulting from the interaction between outflows of multiple film cooling holes, we provide the detail of the flow fields from various view angles for further analysis.

In Figure 8, we show the flow fields accompanied with the contours of the particle concentration *c* in the x−z cross sections located in different positions along the s1 and s2 (Figure 8e,f) with F=0.02 (Figure 8a,c) and F=0.08 (Figure 8b,d). Here, we denote the cross section at position 1 along s1 when F=0.02 as “F=0.02−s1−1” for the convenience of description. We highlight the sections in which the cooling outflow occurs, and the generated vortices by cooling air.

In “F=0.02−s1−1”, we observe a high concentration of particles near the wall and upward flow of the fluid. The former one may be caused by the small velocity in the boundary layer, which induces the accumulation of the particles, and the latter one can be attributed to the coolant outflow in the downstream. In “F=0.02−s1−2” and “F=0.02−s1−3”, we observe the flow in transverse direction that is caused by the cooling outflow from the neighboring cooling holes, namely, H1 and H2. In “F=0.02−s1−4”, the outflow of the cooling gas significantly disturbs the distribution of the particle concentration in the boundary layer, and generates a pair of “kidney” shape vortices (clearly seen in “F=0.08−s1−4”) in the downstream. This pair of vortices moves upward and induces another pair of vortices underneath in the downstream (as shown in “F=0.02−s1−5” and “F=0.02−s1−6”). Although these flow field variations have been discovered in many studies of film cooling, they illustrate the strong mixing near the wall caused by the cooling outflow. In Figure 8c,d, we observe a similar variation of flow field along s2 after two film cooling holes, but the concentration contours change by the interaction between the two cooling outflows, particularly shown in “F=0.08−s2−2” and “F=0.08−s2−6”. The mixing caused by the outflow from H2 induces lower particle concentration in its tail region, which is further diluted by the cooling outflow from H7, resulting in a region with nearly zero particle concentration in the tail region (shown in “F=0.08−s2−7”).

In Figure 9, we present the velocity field and concentration contour in y−z cross sections along s1 and s2. Along s1, the incoming boundary layer flow with high particle concentration is lifted and diluted by the cooling air (Figure 9a,b). Along s2, we observe that the diluted flow by the cooling air from the first hole is further mixed with the cooling air from the second hole. In particular, a region without any particle appears behind the second hole as shown in Figure 9d. In Figure 9e,f, the streamlines of the cooling air originating from two cooling holes are respectively colored with grey and black to illustrate the interaction between the cooling flows from two holes along s2. The outflow from the first hole, overwhelmed by the flow in the mainstream, enters the boundary layer, and is further lifted by the outflow from the second hole.

Based on these flow fields and concentration contours, we are able to understand the particle deposition characteristics under the interactions between the cooling outflows from the neighboring cooling holes. For the film cooling outflow, due to its low velocity, it enters into the boundary layer of the mainstream and induces the mixing in the mainstream, which not only dilutes the particle concentration in the boundary layer but also gives higher possibility of the particle deposition by the induced disturbance. Therefore, the film cooling outflow from one hole may not necessarily eliminate the particle deposition in its tail region (Figure 6a). The cooling outflow from the second hole lifts the diluted flow after the first one and supplies the boundary layer with a flow of zero particle concentration, leading to a tail region without particle deposition (Figure 6a). For the particle deposition in between these tail regions, it can be explained by the redistribution of the particles in the mainstream under the disturbance of the cooling flow, as shown in Figure 10. The particle concentration in the x−z cross section is uniform along the *x*-axis before the film cooling holes (y=−15 mm) and it becomes uneven after the film cooling holes (y=1.5 mm). The generated vortices induces the tail regions with lower particle concentration and those in between with higher particle concentration (Figure 10c,d), resulting in more particle deposition in areas between the tail regions (Figure 4a).

## 4. Conclusions

In conclusion, the particle deposition in the vicinity of multiple film cooling holes are presented by simulation and validated by experiment. In general, the supply of cooling air through the hole reduces the particle deposition, and this effect increases with the blowing ratio. The interactions between the outflows of the film cooling holes reveals different deposition characteristics in different regions of the surface. The particle deposition occurs after the hole in the first row, but it can be eliminated in the tail regions of the holes starting from the second row. The flow field analysis from simulation indicates that the interaction between the outflows of different holes generates such characteristics of particle deposition. Due to the high particle concentration in the boundary layer near the wall, the cooling air from the hole in the first row dilutes the concentration, but induces the disturbance that facilitates particle deposition. The cooling air from the hole in the second row lifts the diluted flow and separates the particles from the wall, generating a tail region without deposition. These revealed particle deposition characteristics could be helpful in understanding the deposition mechanism in practical applications and benefit the design of the film cooling and transpiration cooling.

## Figures and Tables

**Figure 1 micromachines-13-00523-f001:**
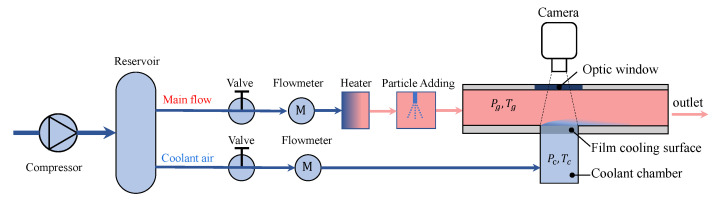
Schematic of the experiment apparatus for particle deposition experiment.

**Figure 2 micromachines-13-00523-f002:**
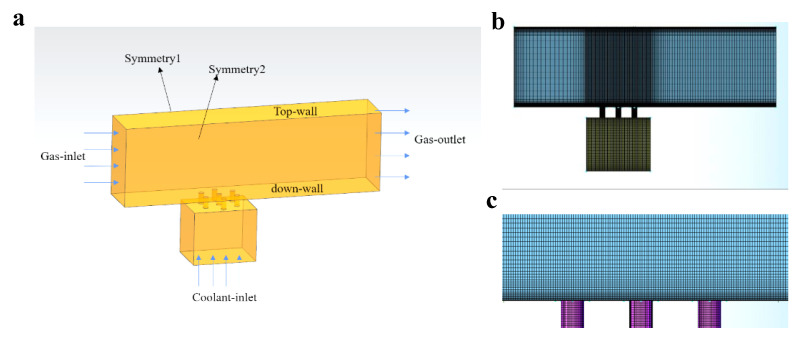
(**a**) Schematic of the simulation model and the boundary conditions. The detail of the generated mesh from the side view (**b**) and in the near wall region (**c**).

**Figure 3 micromachines-13-00523-f003:**
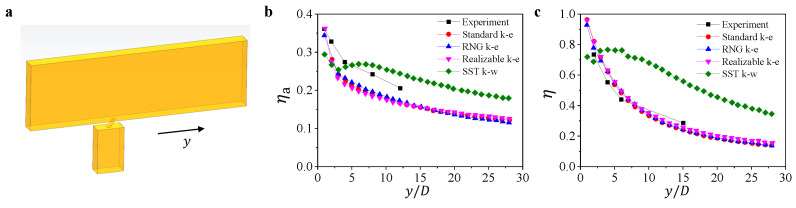
(**a**) Schematic of the simulation for model validation. The variation of (**b**) the averaged cooling efficiency $\eta_{a}$ in transverse direction and (**c**) the cooling efficiency η in the center line along the flow direction for simulation result and experiment data from reference [26].

**Figure 4 micromachines-13-00523-f004:**
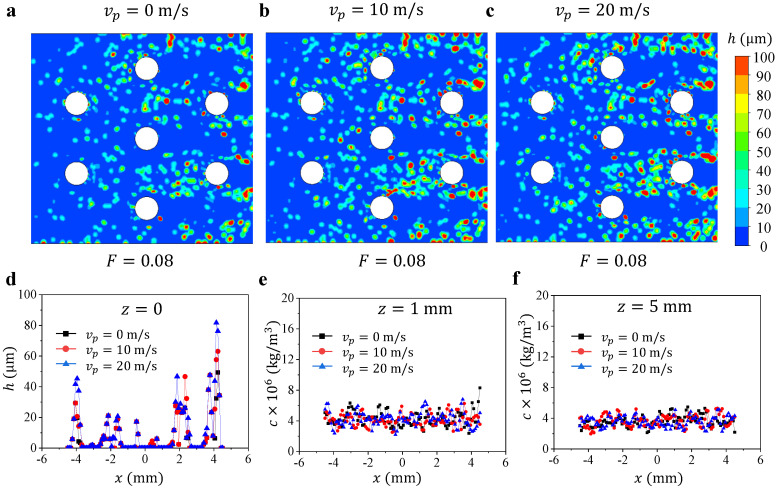
Contours of particle deposition thickness *h* at F=0.08 with different $V_{p}$: (**a**) $V_{p}=0$ m/s, (**b**) $V_{p}=10$ m/s, and (**c**) $V_{p}=20$ m/s. (**d**) The deposition thickness *h* along *x* direction on the surface at y=1.5 mm. The particle mass concentration *c* along *x* direction at (**e**) (z=1 mm, y=1.5 mm) and (**f**) (z=5 mm, y=1.5 mm).

**Figure 5 micromachines-13-00523-f005:**
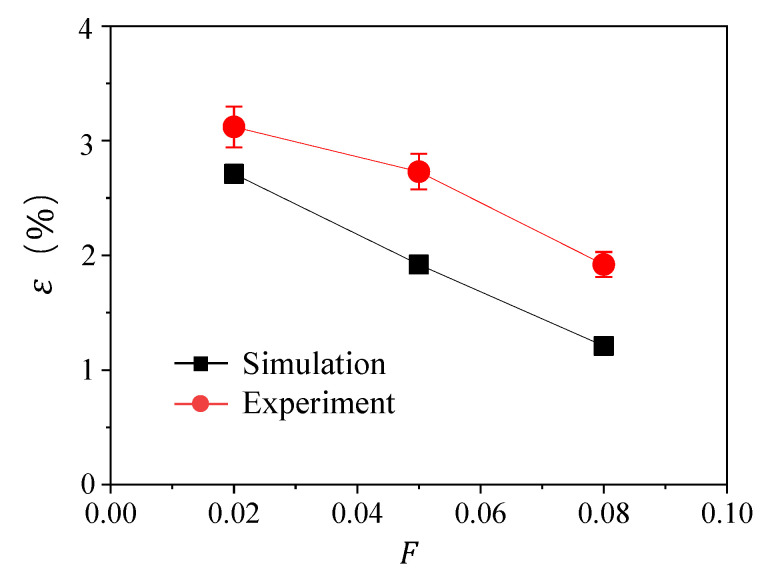
The variation of particle capture efficiency ε with blowing ratio *F* for experiment result and simulation data.

**Figure 6 micromachines-13-00523-f006:**
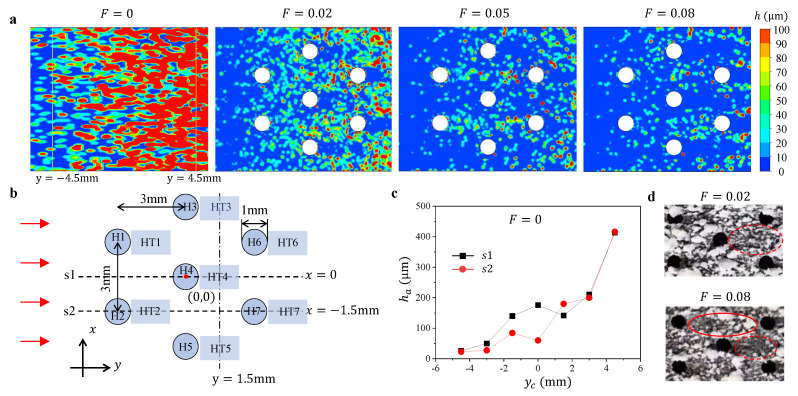
(**a**) Contours of particle deposition thickness *h* with different blowing ratio *F*. (**b**) Schematic of the film cooling holes arrangement and the labeled areas. (**c**) The variation of averaged deposition thickness $h_{a}$ with the *y* coordinate $y_{c}$ of the center point of the chosen areas along s1 and s2. (**d**) Experiment snapshots of the particle deposition in the middle of the flat surface at F=0.02 and F=0.08.

**Figure 7 micromachines-13-00523-f007:**
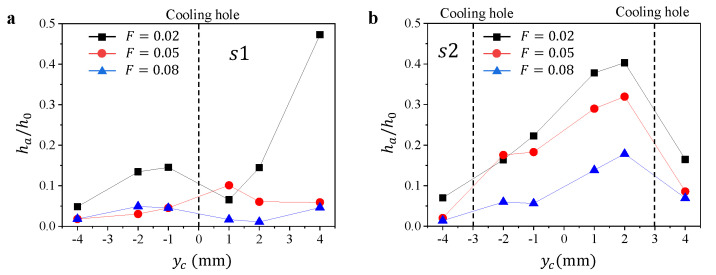
The variation of normalized thickness $h_{a}/h_{0}$ with $y_{c}$ for different blowing ratio *F* along (**a**) s1 and (**b**) s2.

**Figure 8 micromachines-13-00523-f008:**
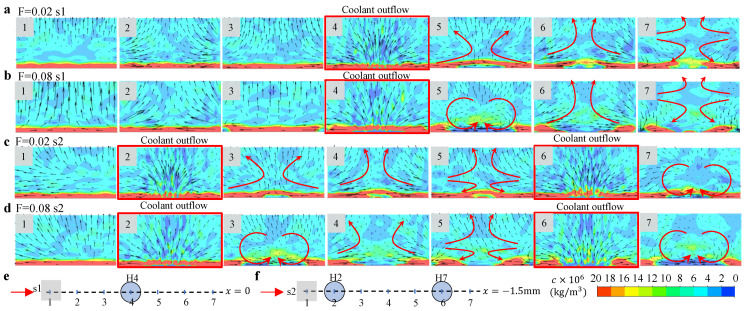
The contours of the particle mass concentration *c* with accompanied velocity vector field at different x−z cross sections along s1 or s2 at different blowing ratios: (**a**) F=0.02-s1, (**b**) F=0.08-s1, (**c**) F=0.02-s2, and (**d**) F=0.08-s2. The cross sections where cooling outflow happens are highlighted. The schematics of the positions of the cross section along (**e**) s1 and (**f**) s2.

**Figure 9 micromachines-13-00523-f009:**
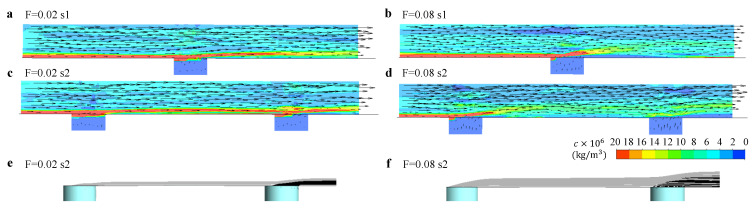
The velocity field and concentration contour in y−z cross section along s1 or s2 at different blowing ratios: (**a**) F=0.02-s1, (**b**) F=0.08-s1, (**c**) F=0.02-s2, and (**d**) F=0.08-s2. The streamline of the cooling air originated from two cooling holes along s2 for (**e**) F=0.02 and (**f**) F=0.08. The streamline from the first hole is colored in grey, while that from second one is colored in black.

**Figure 10 micromachines-13-00523-f010:**
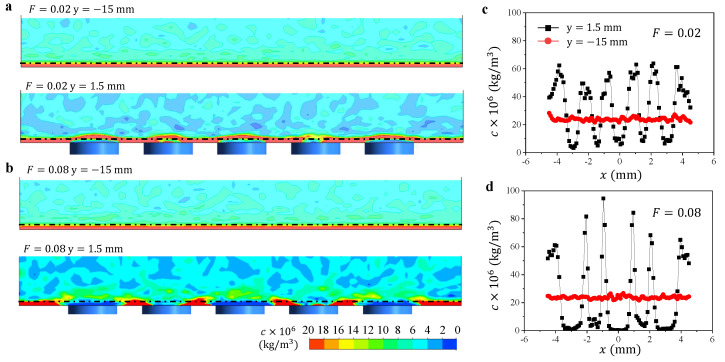
The contours of the particle concentration *c* in the x−z cross section before the hole (y=−15 mm, upper panel) and after the hole (y=1.5 mm, lower panel) for (**a**) F=0.02 and (**b**) F=0.08. The variation of particle concentration *c* with *x* along the dash-dot lines (z=0.5 mm) in (**a**) and (**b**) for (**c**) F=0.02 and (**d**) F=0.08.

**Table 1 micromachines-13-00523-t001:** Physical properties of the added wax particle.

Particle Property	Value
Size range	5–40 μm
Density	900 kg/m^3^
Specific heat	984 J/(kg·K)
Thermal conductivity	0.5 W/(m·K)

## Data Availability

All processed data in this study are included in this published article. Raw data will be provided on request from the corresponding author.

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
