# Peer review of "Particle Deposition in the Vicinity of Multiple Film Cooling Holes"

_micromachines, 2022, doi:10.3390/mi13040523_

Round 1

Reviewer 1 Report

In my opinion, the article is very well prepared, both substantively and graphically. The experiments / models are thoroughly described and presented. The drawings and diagrams are accurately described. The research results are interesting. In my opinion, I only propose to consider the development of the literature.

The article focuses on the research on the efficiency of turbine blades cooling and protection against the deposition of pollutants particles on them. The article presents experimental and model studies of particle deposition in the vicinity of multiple film cooling holes. The introduction contains a good literature recognition of the issue. The description of the research is presented clearly. At the beginning of the article, I propose to insert a list of symbols.

Author Response

We thank the Reviewer for the support of our manuscript. We have added a nomenclature, listing the used variables and their definitions, in the end of the main text.

Reviewer 2 Report

The work is very interesting and seems to be an original. The paper is well written with a good discussion of the results. The validation of the calculation code gave a good agreement with the experimental results.  I enlist them below:

Authors are suggested to consider different shaped Film Cooling Holes in the present studies.

  • What problem was studied and why is it impact? What is the novelty of the work and where does it go to beyond previous effects in the literature
  • Explain Particle deposition with respect to industrial applications.

  • You should add appropriate references for governing equation.
  • How select the range of the variable parameters?
  • Explain How to develop Mathematical formulation and provide all details which term will be not considered from those equations.
  • The main weakness of this paper is they have not provided the application of the present study in the real world problem and are suggested to provide.
  • Revise the abstract section to accurately present not only the aim of the study and methodology

Author Response

Response to Reviewer 2

We thank the Reviewer for the support of our manuscript. In following, we answer the questions in the order that they appear in the review.

Question 1: Authors are suggested to consider different shaped Film Cooling Holes in the present studies.

Answer: We thank the Reviewer for this helpful comment. Although the effect of film cooling hole shape has already shown its effect on the film cooling efficiency, this may merit another study which we may continue to work on in the future.

Question 2: What problem was studied and why is it impact? What is the novelty of the work and where does it go to beyond previous effects in the literature?

Answer: The problem we were dealing with in this paper is the particle deposition on the turbine blade, which is an evitable engineering issue that harms the working safety of the turbine blade. Therefore, it will be helpful to obtain insights into this phenomenon. For the novelty of this paper, we found most of the studies on particle deposition only focus on the configuration of single film cooling hole or single row of holes, which makes the interaction between outflows from different holes lacking. This is the contribution and novelty of our paper. We numerically illustrate the interactions between outflows in different rows do have a significant effect on the characteristics of particle deposition on the surface. We have added more information in the introduction part.

Question 3: Explain Particle deposition with respect to industrial applications.

Answer: We thank the Reviewer for this comment. We have added more information about this in the introduction part in the main text.

Question 4: You should add appropriate references for governing equation.

Answer: We thank the Reviewer for this comment. We did not include governing equation in the manuscript, but two expressions for the definition of blowing ratio and cooling efficiency. We have added references for these two expressions.

Question 5: How select the range of the variable parameters?

Answer: Our intension is to simulate the flow of transpiration. Therefore, we chose the range of the parameters, including the blowing ratio and particle concentration in the mainstream according to the relevant references and practical applications.

Question 6: Explain How to develop Mathematical formulation and provide all details which term will be not considered from those equations.

Answer: We only have two expressions in our manuscript and have cited relevant papers into the main text.

Question 7: The main weakness of this paper is they have not provided the application of the present study in the real world problem and are suggested to provide.

Answer: We did provide the application of this study in the introduction part. To highlight this part, we have added more information in the main text.

Question 8: Revise the abstract section to accurately present not only the aim of the study and methodology

Answer: We have revised the abstract according to Reviewer’s comment.
